# Correlation between Serum Biomarkers and Lung Ultrasound in COVID-19: An Observational Study

**DOI:** 10.3390/diagnostics14040421

**Published:** 2024-02-14

**Authors:** Amne Mousa, Siebe G. Blok, Dian Karssen, Jurjan Aman, Jouke T. Annema, Harm Jan Bogaard, Peter I. Bonta, Mark E. Haaksma, Micah L. A. Heldeweg, Arthur W. E. Lieveld, Prabath Nanayakkara, Esther J. Nossent, Jasper M. Smit, Marry R. Smit, Alexander P. J. Vlaar, Marcus J. Schultz, Lieuwe D. J. Bos, Frederique Paulus, Pieter R. Tuinman

**Affiliations:** 1Department of Intensive Care Medicine, Amsterdam Cardiovascular Sciences, Amsterdam UMC, Vrije Universiteit Amsterdam, De Boelelaan 1117, 1081 HV Amsterdam, The Netherlandsm.heldeweg@amsterdamumc.nl (M.L.A.H.);; 2Amsterdam Leiden IC Focused Echography (ALIFE, www.alifeofpocus.com), 1081 HV Amsterdam, The Netherlands; 3Department of Intensive Care Medicine, Infection and Immunity, Amsterdam UMC, University of Amsterdam, Meibergdreef 9, 1105 AZ Amsterdam, The Netherlands; 4Department of Pulmonary Medicine, Amsterdam Cardiovascular Sciences, Amsterdam UMC, Vrije Universiteit Amsterdam, De Boelelaan 1117, 1081 HV Amsterdam, The Netherlandse.nossent@amsterdamumc.nl (E.J.N.); 5Section Acute Medicine, Department of Internal Medicine, Amsterdam UMC, Vrije Universiteit Amsterdam, De Boelelaan 1117, 1081 HV Amsterdam, The Netherlands

**Keywords:** lung, ultrasonography, epithelial injury, endothelial dysfunction, immune activation, respiratory failure, COVID-19, SARS-CoV-2

## Abstract

Serum biomarkers and lung ultrasound are important measures for prognostication and treatment allocation in patients with COVID-19. Currently, there is a paucity of studies investigating relationships between serum biomarkers and ultrasonographic biomarkers derived from lung ultrasound. This study aims to assess correlations between serum biomarkers and lung ultrasound findings. This study is a secondary analysis of four prospective observational studies in adult patients with COVID-19. Serum biomarkers included markers of epithelial injury, endothelial dysfunction and immune activation. The primary outcome was the correlation between biomarker concentrations and lung ultrasound score assessed with Pearson’s (r) or Spearman’s (r_s_) correlations. Forty-four patients (67 [41–88] years old, 25% female, 52% ICU patients) were included. GAS6 (r_s_ = 0.39), CRP (r_s_ = 0.42) and SP-D (r_s_ = 0.36) were correlated with lung ultrasound scores. ANG-1 (r_s_ = −0.39) was inversely correlated with lung ultrasound scores. No correlations were found between lung ultrasound score and several other serum biomarkers. In patients with COVID-19, several serum biomarkers of epithelial injury, endothelial dysfunction and immune activation correlated with lung ultrasound findings. The lack of correlations with certain biomarkers could offer opportunities for precise prognostication and targeted therapeutic interventions by integrating these unlinked biomarkers.

## 1. Introduction

Respiratory failure is the hallmark of severe COVID-19 [1]. However, patients often present with distinct clinical trajectories and outcomes. Correct diagnosis, the classification of pathophysiological pathways and the severity of disease in patients with COVID-19 are necessary for determining the best medical and supportive therapy [2,3]. In recent years, there has been a growing emphasis on the identification and utilization of serum biomarkers with therapeutic and prognostic value in patients afflicted by acute hypoxemic respiratory failure (AHRF) [4,5].

These biomarkers encompass not only serum biomarkers, but also encompass ultrasonographic biomarkers, such as those derived from lung ultrasound [6,7,8]. The latter category, possibly more cost effective and readily accessible [9] than serum biomarkers, has garnered increasing attention in the pursuit of accurate prognostication for patients facing this critical medical condition. The advent of lung ultrasound as a biomarker offers a promising avenue for the improved outcome prediction of AHRF, addressing issues of cost and availability associated with serum biomarkers.

Many biomarkers, whether measured in serum or derived from ultrasonographic assessments, exhibit correlations with each other due to their representation of shared biological pathways or underlying pathophysiological processes, among other reasons [10]. COVID-19 exhibits enhanced inflammation, epithelial injury and endothelial dysfunction contributing to lung injury and lung parenchymal changes [9]. Since lung ultrasound can accurately detect these parenchymal changes [11], one would expect serum biomarkers and lung ultrasound findings to be correlated with each other. It is crucial to scrutinize these correlations, as biomarkers that are closely related may not provide additional value when considered in combination. Surprisingly, despite the potential significance of such correlations, there is a paucity of studies investigating these relationships, particularly within the context of AHRF. The limited exploration of biomarker correlations in this specific medical condition underscores the need for comprehensive research in this area to refine prognostic models and enhance patient care. In this aspect, several therapeutic options, such as imatinib, tocilizumab, vilobelimab and corticosteroids, for COVID-19 have pathways that include these biomarkers [12,13].

The objective of this study was to investigate the correlations between previously proposed serum biomarkers and three recently formulated lung ultrasound findings. For our study, we leveraged a comprehensive database [14] containing the data of 24 serum biomarkers and lung ultrasound examinations, all from patients suffering from AHRF due to COVID-19. Our hypothesis posited that all biomarkers, regardless of their origin (serum or lung ultrasound), exhibit substantial correlations with one another. By probing these correlations, our study aspires to shed light on the interplay between serum biomarkers and lung ultrasound findings in the context of AHRF, ultimately contributing to the refinement of prognostic models and therapeutic targets.

## 2. Materials and Methods

This study is a post hoc analysis of four prospective observational studies performed in a tertiary center in Amsterdam, the Netherlands. Systemic serum biomarkers were previously collected as part of the Amsterdam UMC COVID-19 biobank [14]. This study was a prospective cohort study with serial sampling of serum biomarkers to provide insights in temporal changes and prognostic value of these biomarkers in patients with COVID-19. Blood samples were collected from the 23rd of March until the 26th of May 2020. 

Lung ultrasound data were previously collected from the 19th of March to the 30th of May 2020 as part of routine ultrasound at the emergency department, the LUVCT study [6], or intensive care unit (ICU) [7,8]. The LUVCT study was a prospective cohort study to assess and compare the diagnostic accuracy of lung ultrasound and computed tomography in patients with COVID-19. The first ICU study’s aim [7] was to characterize lung ultrasonographic appearance of patients with COVID-19 admitted to the ICU. The second ICU study’s [8] primary aim was to compare lung ultrasound to computed tomography, a tool for monitoring in patients with COVID-19 admitted to the ICU. This post hoc analysis was approved by the local ethics committee (2022.0100). 

Adult (>18 years) patients with a positive SARS-CoV-2 PCR and with clinical symptoms consistent with COVID-19 admitted to the ward or ICU were eligible for inclusion. Patients were included in this study if lung ultrasound examinations and biomarkers were available, with a maximum of 48 h between measurements. Blood samples were collected during admission. Biomarker sets collected closest to the date of lung ultrasound examination were used for analysis. A total of 24 different biomarkers were included in this analysis. Biomarkers were measured using a Luminex platform or ELISA (Table A1). All analyzed serum biomarkers are known to be involved in the pathophysiological process of AHRF and/or COVID-19 [14,15]. Markers included for analyses were angiopoietin-1 (ANG-1), angiopoietin-2 (ANG-2), thrombomodulin, von Willebrand factor A2 (vWF-A2), surfactant protein D (SP-D), soluble receptor for advanced glycation end product (sRAGE), complement factors (C3a, C5a, C5b-9), CD14, growth arrest-specific 6 (GAS6), pentraxin-3 (PTX-3), procalcitonin, urokinase-type plasminogen activator receptor (uPAR), vascular cell adhesion molecule 1 (VCAM-1), C-reactive protein (CRP), interleukins (IL-18, IL-1ra, IL-6, IL-8), TNF-R1, TNF-α and bicarbonate. CRP and bicarbonate values were retrieved from the electronic patient file.

Lung ultrasound measurements were performed or supervised by certified physicians at the emergency department or ICU. Patients who received an ultrasound examination at the emergency department but were transferred to the ICU within 24 h were classified as ICU patients. Lung ultrasound measurements were performed with a six- [7] or twelve-region [6,8] protocol. Measurements were performed using a 5–10 MHz linear transducer [7,8] or a 2–5 MHz curvilinear transducer [6]. Offline analysis of lung ultrasound examinations was performed by researchers (MEH, MLAH, AWEL). Loss of aeration and parenchymal abnormalities were semi-quantified using the lung ultrasound score (LUS score). The LUS score per view was defined as normal or <3 B-lines = 0; well-separated B-lines (>2) = 1; coalescent B-lines = 2; lobar consolidation with tissue-like characteristics = 3. LUS score was calculated as the sum of scores of each region (0–36). For each region, the LUS score and presence of pleural abnormalities were determined. Lung ultrasound examinations using a six-region protocol or with missing regions were corrected before analysis using the formula: adjusted score = total score × (36/maximal achievable score). Patients were labeled with the following profiles: 1. A-profile: no sign of aeration loss in anterior or lateral regions (score of 0); 2. B-profile: presence of B-lines (score of 1 or 2) and absence of consolidation (score of 3) in one or more anterior or lateral region(s); or 3. C-profile: consolidation (score of 3) in one or more of the anterior or lateral region(s). Logically, patients labeled with A-profile will have lower LUS scores than patients with C-profile due to the calculation of LUS scores. However, in some patients, presence of B-lines or consolidation does not always result in a higher LUS score. Pleural data were labeled as normal or abnormal. Presence of irregular, thickened and/or small subpleural consolidation was classified as abnormal pleural characteristics. 

Statistical analyses were performed using RStudio version 4.2.1. (R Foundation, Vienna, Austria). Variables were tested for normality using Shapiro–Wilk test, QQ-plots and histograms. Descriptive statistics are presented as mean (±SD), median (interquartile range) or number (%) as appropriate. Statistical significance level was set at a *p*-value ≤ 0.05.

To assess the correlation between biomarker levels and LUS score, either Pearson’s or Spearman’s correlation coefficient with a 95% confidence interval was used, depending on the distribution of variables. Due to the identified difference in pathways between ward and ICU patients with COVID-19 [14], a separate analysis was performed to assess correlation between biomarkers and LUS scores in ward and ICU patients. Correlation was considered to be any of the following:Negligible for correlation coefficients <0.1;Weak for correlation coefficient between 0.10 and 0.39;Moderate for correlation coefficient between 0.4 and 0.69;Strong for correlation coefficient between 0.7 and 0.89;Very strong for correlation coefficient >0.9 [16].

Differences in biomarker levels of patients classified with A, B and C profiles were tested using a Kruskal–Wallis test. Differences in biomarker levels between patients with and without pleural abnormalities were identified with Mann–Whitney-U test. 

No sample size calculation was performed as this study was based on a post hoc analysis.

## 3. Results

A total of 44 patients were included. Population characteristics are shown in Table 1. Timing of lung ultrasound examination and blood withdrawal is shown in Table 2. Four patients (all ward patients) had a positive SARS-CoV-2 PCR without respiratory symptoms or oxygen therapy. Twenty-three patients received a lung ultrasound examination at the emergency department, of which two patients were transferred to the ICU within 24 h. Patients were 1.5 [0–4] days admitted at the ward before being transferred to the ICU. Two patients (5%) were administered prednisone at time of measurements. Three patients (7%) participated in a clinical trial in which they either received imatinib or a placebo at time of measurements. Characteristics of the measured biomarkers and lung ultrasound findings are shown in Table A2.

In the overall cohort, weak correlations were found between serum biomarkers and LUS scores for biomarkers ANG-1, SP-D and GAS6, whereas a moderate correlation was found between CRP and LUS score (*p* < 0.05) (Figure 1). Moderate correlations were also found for ANG-1, GAS6, CRP, procalcitonin and IL-6 in the ward cohort. Furthermore, in ICU patients, a moderate correlation was found for GAS6. All visualizations of the correlations between biomarkers and LUS scores are shown in Figure A1.

Most patients were classified by lung ultrasound as B-profile (70%), followed by C-profile and A-profile with 16% and 14%, respectively (Table A1). C-profile was only present in the ICU cohort. Serum biomarkers which showed a significant difference between patients with A-, B- and C-profiles are shown in Figure 2. All other serum biomarkers are shown in Figure A2.

In 33 (75%) patients, pleural abnormalities in one or more regions of the scanned lungs were found. In two patients, the presence of pleural abnormalities was not reported and lung ultrasound images could not be retrieved from patient medical records. Serum biomarkers which showed a significant difference between patients with normal and abnormal pleural characteristics are shown in Figure 3. All other biomarkers are shown in the Appendix A, Figure A3.

## 4. Discussion

In this observational study on adult patients with COVID-19, we investigated the correlation between bedside serum biomarkers and lung ultrasound findings. We found that LUS score was correlated with several biomarkers that are known to be correlated with severity of disease and worse outcomes in patients with COVID-19. In the ward cohort, we found multiple correlations between LUS score and markers for endothelial dysfunction and immune activation, whereas in the ICU cohort, only GAS6 seemed to be correlated with LUS score. Also, patients with a B- or C-profiles tend to have higher levels of immune activation compared to patients with the A-profile. Finally, no pathophysiological pathways could be identified using pleural abnormalities as a ultrasonographic biomarker. 

A previous study showed that biomarkers of epithelial injury and endothelial dysfunction were correlated with higher LUS scores in patients with acute respiratory distress syndrome (ARDS) [17,18]. Our study shows similar results with a positive correlation for SP-D and a negative correlation for ANG-1 with LUS scores. These biomarkers have been previously described as important markers for the diagnosis and prognosis of patients with ARDS and AHRF [19,20]. In addition, patients with COVID-19 who have increased SP-D levels benefit from imatinib [21,22]. Also, LUS scores and lung profiles showed correlations with IL-6. Anti-IL-6-receptor therapy such as tocilizumab in patients with COVID-19 with high levels of IL-6 has shown to be beneficial by potentially blocking the inflammatory cascade [23,24]. In addition to patient selection to experience the beneficial effects of tocilizumab, the timing of administration is crucial. Administering tocilizumab early to prevent a cytokine storm appears to be essential for maximizing effectiveness [25]. Since lung ultrasound can offer clinicians real-time insights at the bedside and shows correlations with these serum biomarkers, lung ultrasound could facilitate prompt treatment allocation. A recent study showed that among ICU patients with COVID-19, administration of imatinib decreased extravascular lung water when IL-6, TNF-R1 and SP-D levels are increased [22]. This study showed that therapeutic strategies, e.g., therapies targeting endothelial barrier dysfunction, should be tailored to patient characteristics. Since lung ultrasound is an accurate tool to identify extravascular lung water and monitor the involvement of lung aeration [26], it might be a new biomarker to identify different subphenotypes. The identification of subphenotypes based on ultrasonographic biomarkers is yet to be explored.

Interestingly, several serum biomarkers that were previously reported to be sufficient markers of the severity of disease in COVID-19 and are used as therapeutic targets [24,27,28] were not correlated with LUS scores, lung profiles or pleural characteristics. The lack of correlations with these serum biomarkers underscores the potential of combining different measures for prognostication and therapeutic targeting. The contradictory results indicate that ultrasonographic and serum biomarkers show different pathways involved in pathophysiological processes in patients with AHRF. However, prognostic and therapeutic consequences should be further investigated before clinical adaptation of these results. 

In our study, certain serum biomarkers revealed negligible correlations with lung ultrasound findings. These poor correlations between serum biomarkers and ultrasonographic biomarkers such as lung ultrasound as markers for inflammatory lung injury and endothelial dysfunction are in line with previous studies [17,29]. These results suggest a potential limitation in relying solely on these serum biomarkers to replace the diagnostic utility of lung ultrasound. While this may be perceived as a less favorable outcome, the potential for improved diagnostic measures lies in recognizing the complementary role of serum biomarkers when integrated with lung ultrasound biomarkers. This synergy has the potential to enhance prognostication accuracy, underscoring the importance of a comprehensive approach that harnesses the strengths of both modalities for a more nuanced and clinically relevant assessment [30].

This study has several strengths. We assessed the relation between ultrasonographic biomarkers derived from lung ultrasound and a wide variety of serum biomarkers, whereas previous studies focused solely on immune activation biomarkers. In addition, as a result of including both ultrasound examinations on the ward and the ICU, we included patients with varying severity of disease and therefore a wide range of lung ultrasound and biomarkers profiles were analyzed. In addition, we performed separate analyses on ward and ICU patients because different pathways might be involved depending on COVID-19 disease severity. Furthermore, due to frequent blood withdrawals, we limited the time difference between lung ultrasound examination and biomarker profiles, making the results more reliable to actual disease profiles. 

This study also has several limitations. First, this study uses systemic serum biomarkers, whereas the ultrasound examinations are limited to the lung. Therefore, it would have been of additional value to analyze local biomarker levels in the lung. Second, both lung ultrasound findings and biomarker levels change over time. Therefore, future studies should take this into account and evaluate longitudinal data to assess the relationship between lung ultrasound and serum biomarkers. Third, our study did not take any comorbidities or other ongoing pathophysiological processes such as pulmonary of systematic diseases into account for the analysis due to limited sample size. However, these diseases were not highly prevalent in our cohort and are therefore unlikely to have influenced the results. Lastly, although our study has a sample size similar to other studies researching the correlation between biomarkers and lung ultrasound, the results of this study should be interpreted with caution. Larger prospective studies in both ward and ICU patients should be performed to confirm our results.

## 5. Conclusions

In conclusion, this study provides new insights into the correlation between serum biomarkers of epithelial damage, endothelial dysfunction and immune activation, and lung ultrasound findings, including LUS scores, lung profiles and pleural characteristics, in patients with COVID-19. LUS scores show correlation with ANG-1, GAS6, CRP and SP-D. In addition, we found that patients with a B- or C-profile had higher levels of immune activation. Further research is needed to assess the value of lung ultrasound combined with serum biomarkers for treatment allocation.

## Figures and Tables

**Figure 1 diagnostics-14-00421-f001:**
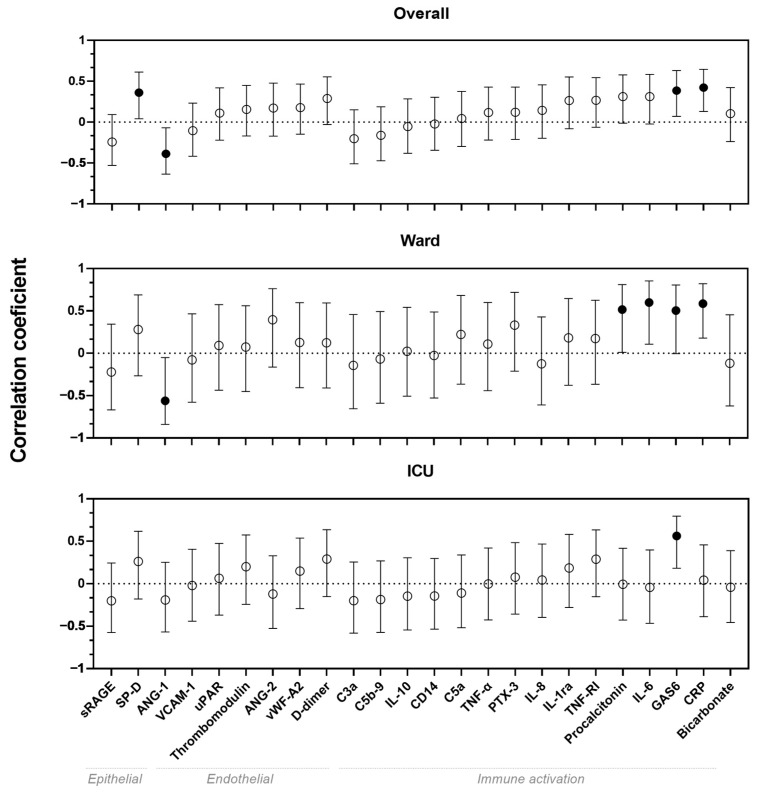
Correlation coefficients with 95% confidence intervals between lung ultrasound score and biomarkers for the overall cohort, ward patients and ICU patients. Black dots indicate statistical significant correlation coefficient (*p* < 0.05). ANG: Angiopoietin. C: Complement component. CD: Cluster of differentiation. CRP: C-reactive protein. GAS: Growth arrest specific. IL: Interleukin. PTX-3: Pentraxin-3. sRAGE: Soluble receptor for advanced glycation. SP-D: Surfactant protein-D. TNF: Tumor necrosis factor. uPAR: Urokinase-type plasminogen activator receptor. VCAM-1: Vascular cell adhesion protein 1. vWF: von Willebrand factor.

**Figure 2 diagnostics-14-00421-f002:**
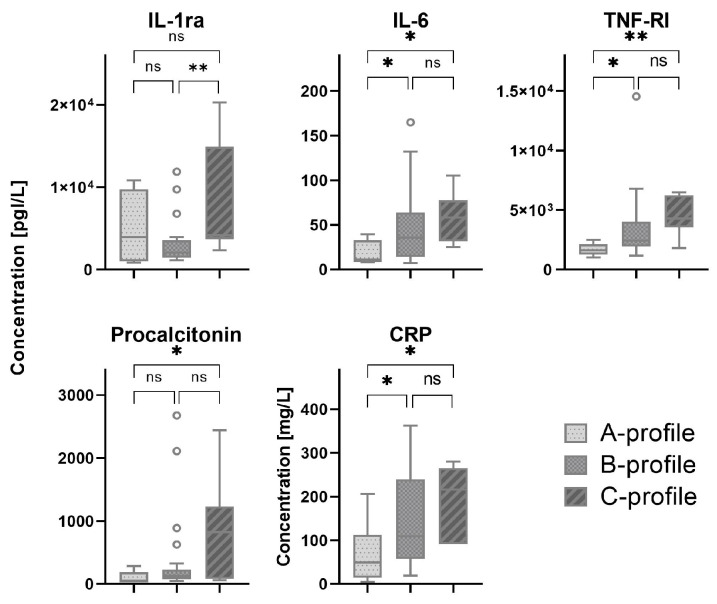
Biomarker levels in patients with different lung ultrasound profiles. The white circles depict the outliers. * indicates *p*-value < 0.05, ** indicates *p*-value < 0.01, ns indicates not significant. CRP: C-reactive protein; IL: interleukin; TNF: tumor necrosis factor.

**Figure 3 diagnostics-14-00421-f003:**
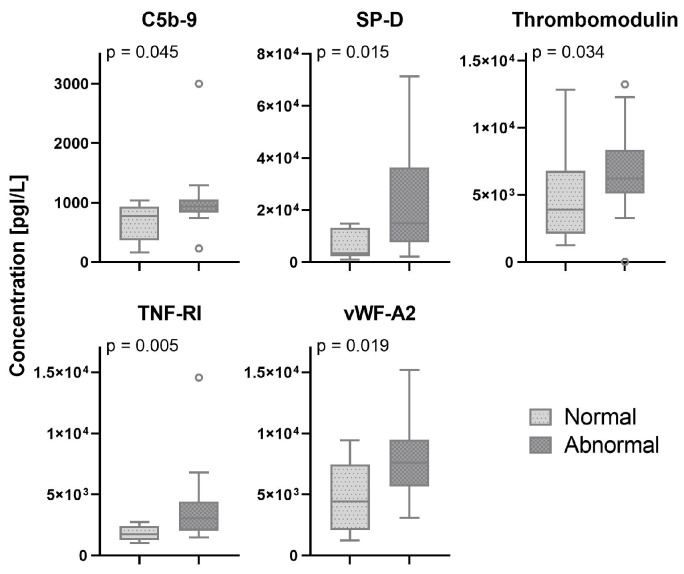
Biomarker levels in patients with normal and abnormal pleural characteristics. The white circles depict the outliers. D: Complement component. SP-D: Surfactant protein-D. TNF: tumor necrosis factor; vWF: von Willebrand factor.

**Table 1 diagnostics-14-00421-t001:** Population characteristics.

Characteristic	Overall (*n* = 44)	Ward (*n* = 21)	ICU (*n* = 23)
Age—years	67 [41–88]	67 [41–88]	67 [45–78]
Female sex	11 (25)	7 (33)	4 (17)
Body mass index Missing	28.1 [22.4–39.2] 10 (23)	27.0 [23.6–37.8] 7 (33)	28.6 [22.4–39.2] 3 (13)
**Medical history**			
Hypertension	18 (41)	9 (43)	9 (39)
Diabetes mellitus	12 (27)	6 (29)	6 (26)
COPD	7 (16)	4 (19)	3 (13)
Usage of immunosuppressive agents prior to admission	7 (16)	1 (5)	6 (26)
**Laboratory values at admission**			
Glucose—mmol/L Missing	8.2 (±2.5) 3 (7)	7.1 [6.0–8.0] 3 (14)	7.8 [7.2–9.7] -
White blood cell count—×10^9^/L	7.4 [4.8–10.0]	6.0 [4.4–7.5]	8.9 [6.2–12.9]
Lymphocytes—×10^9^/L Missing	0.9 [0.6–1.3] 1 (2)	0.81 [0.58–1.25] -	0.91 [0.74–1.32] 1 (4)
Platelets—×10^9^/L	240 [161–328]	205 [149–311]	275 [169–389]
Hemoglobin—mmol/L	8.0 [7.1–8.5]	8.3 [7.6–8.8]	7.6 [6.1–8.3]
Creatinine—µmol/L	91 (±45)	85 [58–101]	81 [65–100]
CT severity score Missing	15 [12–18] 5 (11)	11 [7–14] 5 (24)	18 [15–20]
COVID-19 Reporting and Data System (CO-RADS)	5 [5–6]	5 [4–5]	6 [5–6]
Sequential Organ Failure Assessment (SOFA)	-	-	9 [2–13]
Modified early warning score (MEWS)	-	2 [0–5]	-
Mechanically ventilated during LUS examination	18 (41)	-	18 (78)

Values are mean (±SD), median [IQR] or *n* (%) as appropriate. ICU: intensive care unit. LUS: lung ultrasound.

**Table 2 diagnostics-14-00421-t002:** Timing of lung ultrasound examination and blood withdrawal.

Characteristic	Overall (*n* = 44)	Ward (*n* = 21)	ICU (*n* = 23)
**Time until LUS examination**			
Symptoms—days Missing	10 [0–45] 4 (9)	7 [1–45] 3 (14)	14 [0–43] 1 (4)
Hospital admission—days	-	0 [0–2]	-
ICU admission—days	-	-	6 [0–37]
Mechanical ventilation—days	-	-	6 [1–24]
Time between blood withdrawal and LUS examination—days	1 [0–2]	1 [0–2]	1 [0–1.5]

Values are median [IQR] or *n* (%) as appropriate. ICU: Intensive care unit; LUS: lung ultrasound.

## Data Availability

The data that support the findings of this study are available from the corresponding author upon reasonable request.

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
