# Peer review of "Correlation between Serum Biomarkers and Lung Ultrasound in COVID-19: An Observational Study"

_diagnostics, 2024, doi:10.3390/diagnostics14040421_

Round 1
Reviewer 1 Report
Comments and Suggestions for Authors
Dear authors,
I read with interest the article ‘Correlation between serum biomarkers and lung ultrasound in COVID-19 pneumonia: an observational study’ authored by Mousa et al. Although the study provides insights into possible connections between serum biomarkers and lung ultrasonography results in COVID-19 patients, it is essential to analyse the findings with a discerning viewpoint. There are several major factors that need to be taken into account:
1. Limited Sample Size and Diversity:
The study included only 44 patients, and the demographics may not be representative of the entire COVID-19 patient population. The small sample size raises questions regarding the applicability of the findings, and it is crucial to carry out bigger, more varied investigations to confirm the indicated connections.
2. Observational Nature of the Study:
The study is observational in nature, making it challenging to establish a cause-and-effect relationship between serum biomarkers and lung ultrasound findings. Confounding factors may influence the observed correlations, and additional controlled experiments or longitudinal studies are needed to establish a more robust causal relationship.
3. Selection Bias and ICU Patients:
The study included a significant proportion of ICU patients (52%), potentially introducing selection bias. Patients in the ICU may have more severe disease, and their biomarker profiles might not be representative of those with milder cases. This could impact the generalizability of the findings to the broader spectrum of COVID-19 patients.
4. Clinical Relevance of Correlations:
While statistically significant correlations were reported, the clinical significance of these correlations remains uncertain. It is crucial to determine whether the observed correlations have practical implications for patient management, prognosis, or treatment allocation. Please address these facets inside the Discussion section. Furthermore, I strongly advocate for enhancing the Discussion section by incorporating additional comparisons from other studies. Additionally, the desired number of references should be approximately 30.
5. Incomplete Exploration of Biomarkers:
The study did not find correlations with several serum biomarkers. The reasons for these non-correlations should be explored further. Understanding why certain biomarkers did not show correlations could provide insights into the complex pathophysiology of COVID-19.
6. Minor points:
· Please substitute the term 'radiographic biomarkers' with either 'imagistic' or 'ultrasound';
· In lines 130-133, I recommend presenting the information in either a bulleted or numbered list;
· Table 1: There is one feature that does not have any variables - ‘Time until LUS examination – days’;
· Kindly present the conclusion as a distinct section labelled "5. Conclusion". Conclude your results instead of making a generic statement.;
· Additionally, some moderate editing of the English language is necessary.
In conclusion, while the study presents intriguing correlations between serum biomarkers and lung ultrasound findings in COVID-19 patients, it is essential to interpret the results cautiously. Further research with larger sample sizes, diverse populations, and rigorous experimental designs is warranted to strengthen the evidence base and translate these findings into clinically meaningful applications.

Moderate editing of the English language is necessary.
Table 1: There is one feature that does not have any variables - ‘Time until LUS examination – days’
Author Response
We thank the reviewer for taking the time to review our manuscript. Please find the detailed responses in the attachment.

Reviewer 2 Report
Comments and Suggestions for Authors
Comments:
1. Clinical characterization of patients is not sufficient: what was the volume of pneumonia according to CT scan? General clinical laboratory data are missing: general blood count, biochemical analysis: glucose levels, etc.
2 Although the title of the article contains pneumonia, the text of the article itself says nothing about pneumonia. It would be useful to expand the clinical description of pneumonia in patients.
3. Inclusion criteria should be described in more detail, including patient diagnoses. Exclusion criteria should also be written, as comorbidities may have affected the results of the study.
4. Did the patients undergo radiologic examination (CT)?
Author Response

(The authors gave the same response as above.)

Reviewer 3 Report
Comments and Suggestions for Authors
The current study titled “Correlation between serum biomarkers and lung ultrasound in COVID-19 pneumonia: an observational study” Ref: 2839441 deals with an important subject. The study is a part of a serial studies dealing with COVID-19 observations due to serum biomarkers and radiographic biomarkers. Determination of correlation(s) between serum biomarkers (24 different biomarkers were utilized in the current study) and the COVID-19 infection through cost effective lung ultrasound can assist in the therapeutical impact. 44 patients were considered in the current study (ward = 21, ICU = 23). Mild to weak observations for this study were exhibited however, it has been recommended that a future study should consider only the local lung biomarkers to optimize better correlations for this disease. Few revisions are needed for this study.
- Prospection for future studies recommendation should mention the specific biomarkers that should be studied. This is useful for many research group(s) interested in this subject, can save time and money needed for analyzing huge number of unneeded biomarkers.
- List of abbreviations considered is needed.
Author Response

(The authors gave the same response as above.)

Round 2
Reviewer 1 Report
Comments and Suggestions for Authors
Dear authors,
I read with interest the revised version of the article ‘Correlation between serum biomarkers and lung ultrasound in COVID-19 pneumonia: an observational study’ authored by Mousa et al. The study provides insights into possible connections between serum biomarkers and lung ultrasonography results in COVID-19 patients.
I would like to express my appreciation to the authors for considering my suggestions during the article review process. However, I would like to bring attention to two issues that were not addressed.
1. Table 1: There is one feature that does not have any variables: ‘Time until LUS examination - days’
2. The desired number of references should be approximately 30. Here I would like to share more information about the paper criteria of Diagnostics Journal: *Original research articles*: more than 4000 words in main body; around or more than 30 references; at least five tables or figures with detailed experimental results are recommended.
Please modify the aforementioned aspects accordingly.
Comments on the Quality of English LanguageMinor revisions to the English language are necessary. Please review the text once more and make the necessary corrections.
Author Response
Thank you very much again for taking the time to review our manuscript. Please find the detailed responses below and the revised manuscript in the attachment. Changes are highlighted in yellow.
- Table 1: There is one feature that does not have any variables: ‘Time until LUS examination - days’
Thank you for this comment. ‘Time until LUS examination –days’ is a heading for the variables below, similarly to ‘laboratory values at admission’ and ‘medical history’. We made a separate table with the timing features and moved ‘Time between blood withdrawal and LUS examination –days’ for more clarity.
- The desired number of references should be approximately 30. Here I would like to share more information about the paper criteria of Diagnostics Journal: *Original research articles*: more than 4000 words in main body; around or more than 30 references; at least five tables or figures with detailed experimental results are recommended.
Thank you for your extensive review. We increased our amount of references up to 30, split table 1 into two tables and increased our word count. Furthermore, we are aware of the paper criteria of Diagnostics Journal and the editor also requested to increase our word count to facilitate transparent and open science. We are happy to cooperate in order to create transparent and open science. However, we strongly feel like our manuscript encompasses all the information to reproduce our results and we have provided several figures and tables in the supplementary materials, which we think should not be moved to the main body of the paper in order to maintain clarity for the reader. We think we are therefore complying with the paper criteria of the journal.

Reviewer 2 Report
Comments and Suggestions for Authors
The authors answered some of my questions and added new information to the article, which improved the quality of the article. However, it is necessary to expand the limitations of the study by not analyzing concomitant respiratory diseases such as COPD or interstitial lung disease that may have affected the results of the study, as well as not taking into account other diseases with systemic inflammation that patients may have had. Materials and methods it is recommended to complete the description of reagents used to analyze biomarkers, add the names and models of all instruments.
Author Response
Thank you very much again for taking the time to review our manuscript. Please find the detailed responses below and the revised manuscript in the attachment. Changes are highlighted in yellow.
The authors answered some of my questions and added new information to the article, which improved the quality of the article. However, it is necessary to expand the limitations of the study by not analyzing concomitant respiratory diseases such as COPD or interstitial lung disease that may have affected the results of the study, as well as not taking into account other diseases with systemic inflammation that patients may have had. Materials and methods it is recommended to complete the description of reagents used to analyze biomarkers, add the names and models of all instruments.
Thank you for your review and comment. We expanded the limitations in the discussion, lines 255-259.
In addition, we added the information concerning the biomarkers, line 90 and table A1 in the supplementary materials.
